# Pokémon GO, Went, Gone…—Physical Activity Level, Health Behaviours, and Mental Well-Being of Game Users: A Cross-Sectional Study

**DOI:** 10.3390/healthcare13182334

**Published:** 2025-09-17

**Authors:** Michał Giller, Tomasz Kowal, Wirginia Likus, Anna Brzęk

**Affiliations:** 1Department of Anatomy, Faculty of Health Sciences in Katowice, Medical University of Silesia in Katowice, 40-752 Katowice, Poland; tomasz.kowal@sum.edu.pl (T.K.); wlikus@sum.edu.pl (W.L.); 2Department of Physiotherapy, Faculty of Health Sciences in Katowice, Medical University of Silesia in Katowice, 40-752 Katowice, Poland

**Keywords:** Pokémon GO, physical activity, civilization disease, augmented reality (AR), mobile gaming, smartphone, mental health, health promotion, public health, social interaction, motivation, well-being, IPAQ (International Physical Activity Questionnaire)

## Abstract

Background/Objectives: Pokémon GO has had a substantial global impact, emerging as one of the most prominent smartphone game releases of the 21st century. Beyond its entertainment value, this game has the potential to encourage physical activity alongside recreational engagement. Consequently, it may facilitate the integration of augmented reality into daily routines within the context of advancing mobile device technology. This study aimed to assess the impact of Pokémon GO on users’ physical activity levels, as well as to identify other implicit health outcomes and potential risks. Methods: A cross-sectional study was conducted using the International Physical Activity Questionnaire (IPAQ-long form) and a custom-designed survey (including demographic characteristics) administered to a cohort of 243 Pokémon GO players (112 females and 131 males, mean age 27 ± 7 years). Results: According to IPAQ-long form data, 68% of Pokémon GO users demonstrated high physical activity levels, 29% moderate, and 2.5% insufficient activity. More than 80% of participants reported increased walking distances, and 39% indicated that playing the game had improved their overall mood, while 13% reported enhanced social interactions. However, some findings are concerning, with 27% of respondents admitting to sacrificing sleep, 20% considering themselves addicted, and more than half exceeding the World Health Organization (WHO) screen time guidelines based solely on the time spent playing Pokémon GO. Conclusions: Pokémon GO has a positive impact on users’ physical activity levels, particularly in terms of low-intensity physical activities such as walking. This observational study suggests that the application may be associated with a healthy lifestyle and enhanced interpersonal contact. Prudent and safe usage is advised, as the game has the potential to be addictive and may pose risks when misused.

## 1. Introduction

The widespread adoption of smartphones has transformed global connectivity and functionality, rendering these devices ubiquitous tools for a diverse array of functions. This integration of technology into daily life presents both challenges and opportunities for public health. While increased screen time has exacerbated sedentary behaviour and its associated health risks, technology can also be leveraged to encourage physical activity. Society is increasingly adopting such tools, often motivated by competitive or social trends rather than explicit health-related concerns. These tools hold significant potential in fostering healthier lifestyles by encouraging physical activity and reducing sedentary behaviour [1,2,3,4,5,6].

The World Health Organization (WHO) recommends engaging in 150–300 min of moderate-intensity aerobic activity or 75–150 min of vigorous-intensity activity per week to optimise health outcomes. Adherence to these guidelines is essential for maintaining physiological homeostasis, thereby conferring both physical and mental health benefits. While physical activity is well recognised as a preventive measure, it also offers therapeutic advantages for managing chronic conditions. Regular engagement in physical activity is particularly vital for the physiological, psychological, and social development of children and adolescents, supporting maturation, enhancing cognitive function, and promoting social integration. Consequently, physical activity represents a cost-effective strategy for disease prevention and management, as well as a fundamental component of holistic human development [7,8,9,10].

Augmented reality (AR), defined as the superimposition of virtual elements onto the real world, has gained increasing prominence. As described by Furth, AR provides a real-time, direct or indirect view of a physical environment enhanced by computer-generated information. AR applications have expanded beyond entertainment into various sectors, including sports, healthcare, aviation, the automotive industry, and education, thereby redefining their respective applications [11].

Pokémon GO, a mobile game utilising AR technology, was released in July 2016 for Android and iOS platforms. Developed by Niantic and Nintendo, the game integrates GPS technology to create an interactive experience and has become one of the most widely downloaded games globally. Within its first week, Pokémon GO was installed on 28.5 million devices in the United States and achieved 750 million downloads worldwide within its first year. Inspired by the Pokémon franchise, Pokémon GO is designed to encourage physical activity, primarily walking, which is essential for game progression. Previous research has shown that Pokémon GO mainly promotes light-to-moderate intensity activity. Althoff et al. reported an average increase of approximately 1470 daily steps in the month following installation [12], while Xian et al. observed nearly 1900 additional steps per day [13]. Systematic reviews further confirm that the game’s contribution is largely limited to walking and other incidental movement rather than structured exercise [10,14]. Although Pokémon GO gained widespread popularity, other AR-based health tools, such as Zombies, Run! and Ingress, also aim to promote physical activity through gamified experiences. However, Pokémon GO’s unique appeal may lie in its integration with the established Pokémon franchise and its emphasis on collection and social interaction. According to data from https://ActivePlayer.io, as of August 2025, the game maintained a global monthly active user base of approximately 60 million, indicating sustained popularity and engagement, with a 25% growth in the number of players over the past 12 months [15,16,17].

Given the increasing prevalence of sedentary lifestyles and associated health risks, innovative strategies are essential for promoting physical activity. In the context of growing technological integration, augmented reality (AR) games such as Pokémon GO present a unique opportunity to address this challenge. This study aims to fill a gap in the existing literature by examining the impact of Pokémon GO, a widely popular AR game, on physical activity levels among adult players in Poland. Understanding the potential of such games to promote physical activity, even at low to moderate intensities, is crucial for informing public health interventions and game design strategies aimed at mitigating sedentary behaviour and reducing the risk of lifestyle-related diseases.

### Objectives

The primary objective of this study was to examine the association between Pokémon GO gameplay and physical activity levels among its users in Poland. A secondary aim was to determine whether physical activity among Pokémon GO players adhered to the World Health Organization (WHO) recommendations. Additionally, the study sought to evaluate the correlation between time spent playing Pokémon GO and the WHO guidelines for limiting screen time.

## 2. Materials and Methods

This study employed a cross-sectional design, recruiting a cohort of active Pokémon GO players. The sample comprised 243 individuals (112 females, 131 males), aged 16 to 59 years (mean age: 27 ± 7.0 years). Inclusion criteria required current engagement with Pokémon GO. The study was conducted in accordance with the Declaration of Helsinki and was approved by the Bioethics Committee of the Medical University of Silesia in Katowice (resolution no. BNW/NWN/0052/KB/197/25), dated 8 July 2025. Informed consent was obtained from all participants.

Data collection instruments included the validated Polish version of the International Physical Activity Questionnaire (IPAQ long form), used to quantify participants’ physical activity levels [18]. A systematic meta-analysis across European cohorts confirmed the instrument’s robustness, reporting moderate test–retest reliability (intraclass correlation r_w = 0.74), concurrent validity (r_w = 0.72), and criterion validity (r_w = 0.41) for moderate-to-vigorous physical activity metrics [19]. Nevertheless, as with all self-report tools, limitations related to recall bias and potential misclassification should be acknowledged. Participants were instructed to report the activities that contributed to specific parts of the IPAQ. In part 4, as per the original instructions, they were advised not to include any activities already mentioned; however, some overlap (e.g., walking for transport versus leisure) cannot be entirely excluded.

In addition, a custom-designed survey was administered, consisting of two sections: a demographic questionnaire and a series of questions regarding sports participation and engagement with Pokémon GO. The demographic section collected data on gender, age, and self-reported overall health status. Participants were asked about their involvement in sports, including the type of activity, frequency, duration, and years of participation. The second section focused on participants’ experiences with Pokémon GO, exploring motivations for initiating gameplay, session durations, and perceived changes in physical activity levels, mood, and social interactions related to the game. Our customized questionnaire was developed through a comprehensive and rigorous review of prior research investigating Pokémon GO and health-related behaviours. Its content validity was ensured through iterative consensus among the research team experts and pilot testing for clarity and relevance.

Participants were recruited through convenience sampling via online platforms, including social media groups dedicated to Pokémon GO enthusiasts. Data were collected using a survey administered through Google Forms. The average completion time per respondent was approximately 15 min.

Statistical analyses were performed using Statistica v.13. The Kolmogorov–Smirnov test was used to assess the normality of the distributions. Descriptive statistics, including means, standard deviations, and ranges, were computed. Frequency distributions were calculated, and statistical comparisons and correlations were analysed. In the univariate analysis, dependence tests (R-Pearson, R-Spearman) and tests of significance of differences (t-Student, U-Mann–Whitney, chi-2 Pearson) were used. Statistical significance was assessed at a level of α ≤ 0.05. Given the exploratory and cross-sectional design, with most variables measured on nominal or ordinal scales and psychosocial outcomes captured by single items, the analytical strategy was limited to descriptive and bivariate methods. This approach aligns with common practices in comparable Pokémon GO studies and enables the present findings to be interpreted as hypothesis-generating for future longitudinal research with objective measures.

## 3. Results

The sample comprised 131 males (54%) and 112 females (46%). According to IPAQ data, approximately 68% of participants exhibited high physical activity levels, 29% engaged in moderate activity levels, and 3% demonstrated insufficient activity. Only 2.5% of respondents did not meet the WHO-recommended weekly physical activity guidelines (Table 1).

Overall, males exhibited higher levels of physical activity compared to females, with a difference of 14.5%. The most significant disparities were observed in work-related activities, where males were 27.8% more active, and in household chores, where males were 24% less active. Among males, 49.6% actively participated in sports, compared to 34.8% of females. Women commonly participated in team sports, swimming, and gym workouts, while males predominantly engaged in team sports (Table 2, Figure 1).

Additional gender-related differences were identified in the responses to specific survey items. Men were significantly more likely than women to report playing other mobile games that required outdoor physical activity. Furthermore, trespassing on private property while playing Pokémon GO was more frequently acknowledged by men compared to women, with this difference also reaching statistical significance. These results indicate behavioural divergences between genders that extend beyond overall physical activity levels (Table 3, Appendix A).

Over half of the respondents (53.5%) reported that their primary motivation for playing Pokémon GO was their interest in the Pokémon franchise. A substantial proportion (42.4%) started playing following their peers, while 28.4% began playing to increase their physical activity levels. Other reported motivations included social interaction (16.9%), following social trends (8.6%), interest in Pokémon games (5.3%), and curiosity (5.8%). Additional reasons cited included motivating children to be physically active, diversifying their daily routine, and personal enjoyment (Appendix A).

Nearly all participants (96.7%) reported engaging in Pokémon GO gameplay while walking. A majority (55.6%) also played while using public transportation, and 33.3% while operating a motor vehicle. Additional modes of gameplay included cycling, rollerblading, running, or sedentary gaming (Appendix A).

When asked about the impact of Pokémon GO on their physical activity, 41.6% reported that the game positively influenced their activity levels, while 39.1% noted improvements in their overall mood. Additionally, 13.2% indicated that the game enhanced their social interactions (Appendix A).

The game had minimal impact on participants’ motivation for work or study. A majority (86.4%) reported no effect, while 9.5% noted a decrease in motivation and 4.1% experienced an increase as a result of playing Pokémon GO (Appendix A).

## 4. Discussion

The primary objective of this study was to investigate the physical activity levels among Pokémon GO players, and this aim was successfully achieved. The findings suggest that there is a notable increase in physical activity among respondents, which may be associated with the use of the application. Specifically, 28.4% of participants reported initiating gameplay with the intention of increasing their physical activity, while 80.7% acknowledged an increase in walking frequency linked to Pokémon GO. Furthermore, 72.4% of respondents were motivated to reduce their reliance on motorised transportation when possible, opting to engage with the game en route to their destinations. Additionally, 41.6% of the surveyed population identified Pokémon GO as having the most significant impact on enhancing their physical activity compared to other aspects of their lifestyle (Appendix A).

Existing literature primarily examines the relationship between Pokémon GO gameplay and physical activity, reporting varying degrees of correlation. Studies have documented increases in walking ranging from 15% to 35% [12,20,21], an increase of up to 1.5 times in MET-minutes per week [1,22], and reductions in sedentary behaviour by nearly 50% [22]. Other findings include a daily decrease of 30 min in sedentary time [23] or an increase of 1000 [24] to 2000 [13] steps per day. Notably, active Pokémon GO players demonstrated a 25% increase in walking compared to non-players during app usage [25]. However, these findings are not always consistent, and the magnitude of the effect can vary depending on the study population and methodology.

This study found that participants generally achieved high levels of physical activity, ranging from 3000 to 32,736 MET-minutes per week, regardless of gender. The wide variability in MET-minutes per week highlights differences in physical activity levels among Pokémon GO players and underscores the limitations of using self-reported data to estimate energy expenditure. Since no objective measures of physical activity, such as accelerometers, were employed, we are unable to validate these self-reported values.

The gender-related differences observed in this study are consistent with broader trends in gaming and physical activity. Men accumulated more work-related activity and reported higher engagement in other AR-based games, which may partly explain their greater overall MET values. They were also more likely to report trespassing behaviour, suggesting a higher propensity for risk-taking during gameplay. Similar findings were described by Althoff et al. [12] and Wong [25], who noted that male players tend to engage more intensely with the game and exhibit greater willingness to explore potentially unsafe environments. In contrast, women reported greater contributions from household activities and were more often engaged in individual sports such as swimming or fitness training, aligning with the cultural and social determinants of activity preferences highlighted in previous studies [21,22]. Despite these distinctions, both genders reported comparable improvements in mood and overall physical activity associated with the game, suggesting that its motivational appeal is broadly shared across groups.

In summarising the impact of Pokémon GO on physical activity, the game may serve as a potentially beneficial intervention for enhancing this critical component of health in contemporary life. Given society’s increasing reliance on mobile devices, the app may be associated with effectively promoting health benefits, primarily through low- to moderate-intensity physical activity, predominantly walking. While the reviewed studies consistently demonstrate that Pokémon GO encourages users to wander greater distances and spend extended periods walking, the long-term sustainability of these changes remains unclear. These benefits are most pronounced among previously sedentary individuals, who may experience significant health improvements such as weight loss, although this was not directly assessed in our study. Nevertheless, the findings mentioned above support the assertion that Pokémon GO can effectively increase the physical activity of its users [1,12,13,14,21,23,25,26,27,28,29,30,31,32,33,34,35,36,37,38,39,40,41,42,43,44,45].

A review of the literature indicates that motivation to play Pokémon GO tends to diminish over time. This phenomenon is reflected in a decrease in the number of steps taken after a few weeks compared to initial engagement, as well as shorter gameplay sessions. One possible explanation for this decline is the infrequent updates and limited variety in the app shortly after its launch. However, the current version of Pokémon GO includes numerous innovations and regular weekly and monthly events that help sustain user interest [24,30]. Our cross-sectional design does not allow us to assess the long-term sustainability of Pokémon GO engagement or its impact on physical activity over time. Future longitudinal studies are needed to examine these trends.

A primary objective of this publication was to identify additional health benefits potentially associated with playing Pokémon GO. Both the literature review and survey results suggest that the game positively influences players’ life satisfaction, emotional well-being, and social relationships. However, it is important to note that our outcomes were assessed using single-item questions within a cross-sectional survey, which limits our ability to draw definitive conclusions about causality or the magnitude of these effects. The social potential of the game is evidenced by its capacity to facilitate social interactions among players. This was particularly evident after the June 2017 update, which introduced gameplay elements such as raids. These cooperative events require a specific number of participants to collectively defeat a powerful in-game boss, encouraging players to interact and form groups. As a result, players often expand their social circles and become more sociable, strengthening ties with local communities. Compared to non-players, Pokémon GO users reported reduced negative emotions and stress, which they attributed as reasons for continuing their engagement with the game. Research also suggests that spending time outdoors in green spaces can further alleviate stress [20,22,45,46,47,48,49,50,51]. However, it is important to note that our study did not directly compare Pokémon GO players to a control group of non-players, so we cannot definitively attribute these benefits to the game itself.

Key positive effects of Pokémon GO include enhanced social belonging and improvements in symptoms of depression and anxiety, with greater benefits observed among individuals with higher initial levels of these conditions [52,53,54,55,56,57]. However, these findings are based on correlational studies; therefore, further research using more rigorous methodologies is necessary to better understand these relationships.

A secondary objective of this study raised concerns about screen time associated with Pokémon GO. The World Health Organization (WHO) recommends a maximum of two hours of daily screen exposure. However, the results of this study indicate a roughly equal split between those who adhered to and those who exceeded this limit (Table 1). Notably, time spent playing Pokémon GO does not account for additional phone usage, which may further exacerbate excessive screen time. The literature reports similarly concerning findings, including median playtimes of 2 h per day [13], averages of 3 h per day over five days a week [58], and as much as 4.9 h daily [59]. Given the potential risks, promoting responsible gaming habits is crucial. Game developers could incorporate features such as in-app time caps or reminders to take breaks. Further research is necessary to investigate the long-term effects of excessive screen time on both physical and mental health, as well as to develop strategies for encouraging responsible gaming habits.

Studies also reveal significant risks of gaming addiction. For example, Mejia et al. found that 49% of surveyed adolescents (ages 13–16) played Pokémon GO for more than 2 h daily, with 44% exhibiting internet addiction and 23% showing signs of video game addiction. Players who spent more time on Pokémon GO were more likely to experience addiction symptoms, including adverse parental reactions and increased tardiness [60]. A total of 27.2% of participants in our study reported sacrificing sleep to play the game, aligning with literature estimates of 37.8% [27]. Furthermore, 20.6% of respondents considered themselves addicted to the game, meeting criteria defined by tools such as the Smartphone Addiction Scale (SAS) [61,62]. Behavioural tendencies, such as gaming while walking, were identified as risk factors for habitual and addictive use, especially among younger players [63]. These findings highlight the potential for Pokémon GO to contribute to addictive behaviours, particularly among vulnerable populations. Players must be aware of these risks and take steps to prevent excessive gaming, such as setting time limits for gameplay and prioritising sleep. Future research should investigate the factors that contribute to Pokémon GO addiction and develop interventions to address this issue.

Ashar et al. corroborate these findings, reporting symptoms such as preoccupation with the game, frustration when not playing, and neglect of other activities. They also observed high levels of anxiety (66%) and depression (38%) among players [64]. Kaczmarek highlights the game’s reliance on conditional reward systems, which may stimulate dopamine release similar to gambling, potentially fostering compulsive behaviours [65]. These studies provide further evidence of the potential negative psychological consequences associated with excessive Pokémon GO use. Game developers and public health officials should consider these risks when designing and promoting AR games to ensure their safety and effectiveness.

Pokémon GO also raises safety concerns, particularly when players become overly immersed in the game. In total, 81.5% of respondents in this study did not report pain or discomfort; however, minor injuries have been linked to excessive physical activity performed without ergonomic considerations. Serious risks arise when players lose situational awareness, leading to accidents such as falls, collisions, and traffic incidents. Instances of distracted driving and pedestrian behaviours have been documented, with “smombies” (smartphone zombies) identified as a growing hazard [4,65,66]. The potential for accidents and injuries underscores the importance of promoting safe gaming practices and raising awareness of the risks associated with distracted gameplay. Game developers could incorporate features to enhance situational awareness and prevent players from entering hazardous environments. To further mitigate these risks, geo-fencing could be implemented to restrict gameplay in dangerous areas, such as roadways or bodies of water. Relying solely on warnings during game loading or when exceeding speeds of approximately 30 km/h may prove insufficient.

The potential of Pokémon GO to enhance physical activity levels remains complex. The health benefits of the game are largely incidental—supporting a phenomenon termed “stealth health”—although Pokémon GO was not explicitly designed as a health application, it has influenced players’ health behaviours [14,34]. Literature suggests that younger players tend to lose interest more quickly than older players, failing to sustain increased activity levels [26,40,48,67,68]. Studies also indicate that gameplay intensity is relatively low, with players often stopping to achieve in-game goals [69]. Furthermore, interventions reviewed by Baranowski were insufficient to prevent lifestyle-related diseases [70]. Therefore, while Pokémon GO may offer some benefits for physical activity and well-being, it should not be regarded as a panacea or a substitute for structured exercise or other evidence-based interventions. A balanced approach that incorporates active gaming alongside other forms of physical activity is likely to be most effective for promoting long-term health.

### Limitations

Our study, despite yielding interesting results and offering a broad analytical scope, has several significant limitations that should be considered when interpreting the data and designing future research in this area. One such limitation is the cross-sectional nature of the study, which makes it impossible to establish a clear cause-and-effect relationship between playing Pokémon GO and changes in physical activity levels. It cannot be ruled out that individuals who were already physically active were more inclined to start using the application. Another limitation is the absence of a control group consisting of non-players, which restricts the ability to directly compare the impact of the application with general health and behavioural trends in the broader population. Furthermore, our reliance on self-reported data, collected through the IPAQ-long form (known to over- or underestimate activity) and a custom-designed questionnaire (based on single-item questions), may have introduced bias due to recall errors, social desirability, and differing interpretations of the questions. As the study relied exclusively on questionnaires, the accuracy of the reported physical activity levels and psychosocial outcomes is inherently limited, and objective measures such as accelerometers or validated multi-item scales would have provided stronger validity. Despite these limitations, we believe that our study provides valuable insights into the potential impact of Pokémon GO on physical activity and well-being. However, future research should address these limitations by incorporating more objective measures and rigorous study designs.

## 5. Conclusions

In summary, despite certain risks and limitations, Pokémon GO may serve as a potentially useful adjunct in promoting physical activity—and modestly increasing physical activity levels—and in enhancing mental well-being by reducing stress and fostering social connections, particularly among previously inactive individuals. These findings should be interpreted with caution, as they are based on cross-sectional, self-reported data and therefore cannot establish causality. Rather, they highlight associative patterns that warrant further investigation. While the game may offer incidental health benefits, it also poses risks such as screen overuse, sleep disruption, and physical injury, underscoring the need for moderation and awareness. Considering the limitations outlined above, future studies should employ more diverse and longitudinal research methods, including control groups and objective measurements, to obtain more precise and reliable results. To confirm these preliminary associations, there is a pressing need for long-term studies and interventions to evaluate sustained effects on physical activity, especially within European contexts. Future research should also focus on developing guidelines for the safe and balanced use of AR games to maximise their potential benefits while minimising associated risks.

## Figures and Tables

**Figure 1 healthcare-13-02334-f001:**
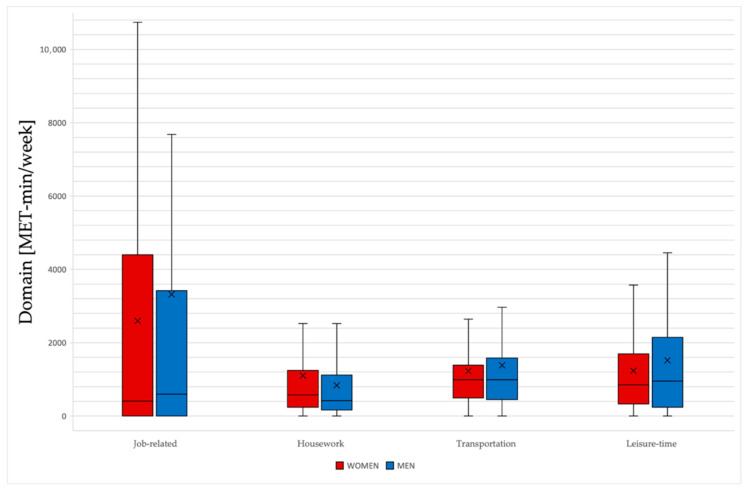
Gender differences in IPAQ-long version domains.

**Table 1 healthcare-13-02334-t001:** Demographics, activity levels and screen time while playing Pokémon GO.

Variables		Women [*n* = 112 (100%)]	Men [*n* = 131 (100%)]
Age (years)	<21	9 (8%)	17 (13%)
	21–40	94 (84%)	109 (83%)
	40<	9 (8%)	5 (4%)
Habitation	village	7 (6%)	13 (10%)
	city, <50,000 inhabitants	7 (6%)	10 (7.5%)
	city, 50–100,000 inhabitants	13 (12%)	10 (7.5%)
	city, 101–200,000 inhabitants	19 (17%)	20 (15%)
	city, >200,000 inhabitants	66 (59%)	78 (60%)
Education	primary	0 (0%)	0 (0%)
	lower secondary	5 (4.5%)	8 (6%)
	technical	3 (2.5%)	3 (2%)
	secondary	33 (29.5%)	43 (33%)
	higher	71 (63.5%)	77 (59%)
Occupation	student	40 (36%)	40 (30.5%)
	unemployed	6 (5%)	5 (4%)
	working	65 (58%)	86 (65.5%)
	retired	1 (1%)	0 (0%)
Health self-assessment	very weak	2 (2%)	0 (0%)
	weak	5 (4.5%)	11 (8%)
	average	35 (31%)	39 (30%)
	good	60 (53.5%)	60 (46%)
	very good	10 (9%)	21 (16%)
IPAQ level	low	3 (3%)	5 (4%)
	moderate	36 (32%)	34 (26%)
	high	73 (65%)	92 (70%)
WHO–activity level	meet criteria	109 (97.5%)	128 (97.5%)
	does not meet criteria	3 (2.5%)	3 (2.5%)
WHO–screen time (playing Pokémon GO)	less than 2 h	55 (49%)	61 (46.5%)
	2 h or more	57 (51%)	70 (53.5%)

**Table 2 healthcare-13-02334-t002:** IPAQ-long version–detailed numbers.

IPAQ Domain	Women		Men		*t*-Student’s Test
	Average ± SD	Range	Average ± SD	Range	
Job-related [MET-min/week]	2594.25 (±4047.59)	0–19,572	3317.19 (±5660.55)	0–27,120	1.156
Housework [MET-min/week]	1102.61 (±1688.92)	0–11,130	838.32 (±1218.11)	0–8190	−1.378
Transportation [MET-min/week]	1229.28 (±1130.87)	0–6930	1384.84 (±1435.87)	0–13,716	0.944
Leisure time [MET-min/week]	1236.43 (±1297.24)	0–5811	1517.71 (±1823.08)	0–13,146	1.399
Total time sitting [min/week]	2803.21 (±1075.69)	540–5880	2760.92 (±1329.92)	405–6720	−0.274
Average time sitting [min/day]	400.46 (±153.67)	77.14–840	394.42 (±189.99)	57.86–960	−0.274
Total score [MET-min/week]	6162.57 (±5451.76)	264–28,386	7058.06 (±7038.81)	148.5–32,736	1.116
Total score [MET-min/day]	880.37 (±778.82)	37.71–4055.14	1008.29 (±1005.56)	21.21–4676.57	1.116

**Table 3 healthcare-13-02334-t003:** Important diverse aspects of playing Pokémon GO.

Question	Answer	Women		Men		Test for Population Proportion
		*n* = 112	100%	*n* = 131	100%	
Do you believe you spend a lot of time playing Pokémon GO?	yes	43	38.5%	58	44%	0.42
	no	69	61.5%	73	56%	0.58
Do you play any other mobile game that requires outdoor physical activity?	yes	6	5%	20	15%	0.11 *
	no	106	95%	111	85%	0.89 *
Do you walk more because of playing Pokémon GO?	yes	90	80%	106	81%	0.81
	no	4	3.5%	5	4%	0.04
Do you use public or private transportation less often if you have the time and opportunity to play Pokémon GO “on the way” to your destination?	yes	82	73%	94	72%	0.72
	no	30	27%	37	28%	0.28
Have you met new friends thanks to playing Pokémon GO?	yes	85	76%	100	76%	0.76
	no	27	24%	31	24%	0.24
Are your social relationships more diverse because of playing Pokémon GO?	yes	70	62.5%	79	60%	0.61
	no	42	37.5%	52	40%	0.39
Have you ever given up any physical activity in favour of Pokémon GO?	yes	2	2%	8	6%	0.04
	no	110	98%	123	94%	0.96
Have you ever felt fatigued because of playing Pokémon GO?	yes	27	24%	41	31%	0.28
	no	60	53.5%	59	45%	0.49
Have you ever felt relaxed thanks to playing Pokémon GO?	yes	84	75%	90	69%	0.72
	no	8	7%	11	8.5%	0.08
Have you ever knowingly trespassed on private property while playing Pokémon GO?	yes	8	7%	26	20%	0.14 *
	no	104	93%	105	80%	0.86 *
Have you ever sacrificed sleep because of playing Pokémon GO?	yes	30	27%	36	27.5%	0.27
	no	82	73%	95	72.5%	0.73
Have you ever found yourself in a health-threatening situation due to inattention caused by playing Pokémon GO?	yes	15	13.5%	19	14.5%	0.14
	no	97	86.5%	112	85.5%	0.86
Do you believe you are addicted to playing Pokémon GO?	yes	17	15%	33	25%	0.21
	no	95	85%	98	75%	0.79

* α ≤ 0.05 for the Test for population proportion.

## Data Availability

The data are stored by the first author in the records of the Department of Anatomy, Faculty of Health Sciences in Katowice, Medical University of Silesia in Katowice.

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
