# Peer review of "Pokémon GO, Went, Gone…—Physical Activity Level, Health Behaviours, and Mental Well-Being of Game Users: A Cross-Sectional Study"

_healthcare, 2025, doi:10.3390/healthcare13182334_

Round 1
Reviewer 1 Report
Comments and Suggestions for Authors
The manuscript offers a compelling and timely exploration of how Pokémon GO influences physical activity and social behavior. The study is well-conceived, with strong methodology and clear presentation of results. I particularly appreciated the multidimensional approach - going beyond physical activity to include screen time, mental well-being, and social interaction.
First, while you do acknowledge the study's limitations, I encourage authors to expand slightly on the implications of using a cross-sectional design without a control group, especially in terms of the limits it places on drawing causal conclusions. Additionally, since your findings touch on the risks of excessive gameplay - such as screen addiction or sleep disruption - it would be helpful to offer a few constructive recommendations or practical considerations for mitigating these risks (for example, in-app time caps or public health messaging). The conclusion is generally well-written but could benefit from a more cautious tone, emphasizing the potential of AR games as useful adjuncts to healthy behavior rather than as primary interventions.
Author Response
Thank you very much for taking the time to review this manuscript. Please find the detailed responses below and the corresponding revisions and corrections marked in red in the updated version of the manuscript.
Comment 1: First, while you do acknowledge the study's limitations, I encourage authors to expand slightly on the implications of using a cross-sectional design without a control group, especially in terms of the limits it places on drawing causal conclusions.
Response 1: We thank the reviewer for highlighting the limitations of our cross-sectional design. We have expanded the discussion of these limitations in the revised manuscript to emphasize the correlational nature of our findings and acknowledge the potential for reverse causality and confounding variables. The addition can be found on page 8, lines 251-253 and 263-265.
Comment 2: Additionally, since your findings touch on the risks of excessive gameplay - such as screen addiction or sleep disruption - it would be helpful to offer a few constructive recommendations or practical considerations for mitigating these risks (for example, in-app time caps or public health messaging).
Response 2: We agree with the Reviewer that practical recommendations are valuable. Accordingly, we have added a new sentence in the Discussion proposing strategies such as in-app time caps, public health messaging, and individual strategies for responsible gaming. The addition can be found on page 8, lines 278-283 and 294-298 (ending on page 9), page 9, lines 304-307 and 314-321.
Comment 3: The conclusion is generally well-written but could benefit from a more cautious tone, emphasizing the potential of AR games as useful adjuncts to healthy behavior rather than as primary interventions.
Response 3: Thank you for this valuable suggestion. We have revised the Discussion and Conclusions section to adopt a more cautious tone, emphasising the role of Pokémon GO and similar AR games as adjuncts to promoting healthy behaviour, rather than primary interventions. The addition can be found on page 9, lines 330-334 and 344-350 (ending on page 10), page 10, lines 352-353.
Reviewer 2 Report
Comments and Suggestions for Authors
Dear Authors,
please find my suggestions in the attached file below.

The manuscript is generally understandable; however, the quality of English could be improved to enhance clarity and precision. Several sections, particularly the Abstract and Discussion, contain awkward phrasing, redundancy, and occasional grammatical errors. Additionally, key concepts such as "physical activity" versus "exercise" are sometimes used interchangeably, leading to confusion. A thorough language revision by a fluent English speaker or professional editor is recommended to improve readability and ensure accurate scientific communication.
Author Response
We greatly appreciate taking the time to review this manuscript and provide such a comprehensive feedback. Please find the detailed responses below and the corresponding revisions and corrections marked in red in the updated version of the manuscript.
Comment 1: The manuscript is generally understandable; however, the quality of English could be improved to enhance clarity and precision. Several sections, particularly the Abstract and Discussion, contain awkward phrasing, redundancy, and occasional grammatical errors. Additionally, key concepts such as "physical activity" versus "exercise" are sometimes used interchangeably, leading to confusion. A thorough language revision by a fluent English speaker or professional editor is recommended to improve readability and ensure accurate scientific communication.
Response 1: We thank the reviewer for pointing out the need for improved language quality. We have carefully reviewed the manuscript and made revisions to enhance clarity and precision. In addition, we have engaged professional editing service to further refine the language throughout the manuscript. Furthermore, we ensured consistent terminology throughout, replacing “exercise” with “physical activity” where appropriate.
Comment 2: The abstract presents a clear overview of the study's aims, methods, and main findings, but it lacks conceptual accuracy in distinguishing between physical activity and structured exercise. It states that Pokémon GO promotes “exercise,” when in fact it encourages unstructured, low-intensity movement, such as walking. This imprecision could mislead readers about the game's healtH impact. The positive outcomes, such as increased ambulation and social interaction, are well-documented. Still, the health benefits are overstated, given the study's cross-sectional design and reliance on self-reported data. Additionally, the mention of risks like addiction and unsafe behawior is commendable but superficially addressed. Overall, the abstract should be revised to clarify the type and intensity of physical activity involved, use more cautious language about health implications, and better reflect the study’s observational nature.
Response 2: We thank the reviewer for their insightful comments on the abstract. We have revised the abstract to clarify the type of physical activity involved (unstructured, low-intensity movement), use more cautious language about health implications, and better reflect the study's observational nature. The addition can be found on page 1, lines 28-29 (in abstract) and thorughout the whole manuscript.
Comment 3: The Introduction addresses a timely topic, using augmented reality games like Pokémon GO to promote physical activity, but presents several conceptual weaknesses. It confuses general physical activity with structured exercise, failing to clarify that the game primarily encourages light, incidental movement (e.g., walking), rather than purposeful exercise as defined by the WHO or the exercise science literature. The background spends excessive space discussing smartphone usage without quickly focusing on the link between AR games and health. Additionally, health benefits are presented too broadly, without distinguishing short-term behavior change from sustained improvements. Prior literature on Pokémon GO and its typical physical activity intensity is also lacking, which weakens the study's justification. While the study objectives are clear, the section would benefit from conceptual precision, a more concise structure, and better integration of relevant evidence to frame the research question properly.
Response 3: We appreciate the reviewer's comments on the introduction. We have excluded term exercise for proper clarification that Pokémon GO primarily encourages light movement like walking, condensed the section of smartphone usage, and incorporated additional literature on Pokémon GO and its typical physical activity intensity. The addition can be found on page 2, lines 70-82.
Comment 4: While the design is clearly described, several limitations weaken its interpretability. First, the exclusive reliance on self-reported measures (IPAQ and survey) introduces potential recall and social desirability bias, especially in a young, tech-savvy sample. It remains unclear how the authors attributed physical activity specifically to Pokémon GO and whether any steps were taken to clean or validate IPAQ data. How did you address possible multiple or overlapping responses in the IPAQ (e.g., walking for transport vs. leisure)? The absence of a control group, even if not necessary considering the chosen design, makes it difficult to assess the actual impact of gameplay compared to general activity trends. Why was no control or comparator group included? Furthermore, the intensity of activity related to the game is not discussed. Did you classify Pokémon Go–related movement as light or moderate? The custom survey lacks any description of pre-testing or psychometric validation. Was this tool piloted or reviewed for reliability? Clarifying these issues is crucial for understanding the strength of the reported associations and for interpreting the public health relevance of your findings.
Response 4: We thank the Reviewer for pointing out this important methodological concern. As clarified in the revised manuscript, our study employed a cross-sectional design based solely on self-reported questionnaires (IPAQ-long form and a custom survey). We did not include a control group, as our primary aim was exploratory—to describe associations between Pokémon GO gameplay and physical activity levels within the studied population. We now explicitly acknowledge that the absence of a control group limits causal inference and the ability to compare results with non-players. This limitation is addressed in the Discussion and further emphasized in the Limitations section. The additions can be found on page 3, lines 109-117, 125-133, page 7, lines 209-213, page 8, lines 245-247, 251-253, 263-265, page 9, lines 344-350 (ending on page 10).
Comment 5: The Results section is clearly structured and presents a detailed account of the sample’s demographics, physical activity levels, and reported experiences with Pokémon GO. The use of IPAQ-derived MET values is appropriate for describing physical activity patterns. Still, the wide variability (e.g., 0–79,740 MET-min/week) raises concerns about outliers or overreporting. Did you apply any exclusion or normalisation procedures for implausible values? While the majority of users reported high activity levels and increased walking, it is not clear how much of this activity was directly induced by gameplay. How did you determine that increases in ambulation were a consequence of playing the game and not due to pre-existing habits? The reporting of positive psychosocial outcomes (e.g., improved mood and social interaction) is valuable, but remains purely self-reported and not linked to any validated scale. Were these outcomes measured with standardised instruments or simple single-item questions? Although gender-based differences in domain-specific physical activity (e.g., job-related or household) are presented, they are not deeply interpreted. Finally, while it is stated that 96% met WHO recommendations, the link between gameplay duration and actual contribution toward those thresholds is not quantified. Can you clarify how much physical activity time was estimated to be related explicitly to Pokémon GO? Overall, the results are rich but would benefit from more precise attribution mechanisms and better alignment between what is measured and what is inferred.
Response 5: We thank the reviewer for raising important questions about the Results section. Most importantly – the point about the variability in MET-min/week values and the potential for outliers. We have identified and corrected errors in our initial data analysis, leading to a completely revised dataset with 112 women and 131 men. As a result of these corrections, we have updated all tables and conducted a completely new statistical analysis to ensure the accuracy of our findings. Following the IPAQ guidelines for data cleaning as well as suggesting our own (mantaining consistent reasoning), we implemented the following exclusion criteria in a sequential manner:
- Excluded records with invalid or inconsistent responses.
- Excluded records reporting less than 10 minutes of a given activity.
- Excluded records reporting 0 days of a given activity, even if minutes were reported (and vice versa).
- Excluded records where the sum of walking, moderate, and vigorous activity exceeded 960 minutes per week (16 hours).
- Excluded records with zero minutes reported for sedentary behavior per day and per week.
- Excluded records where sedentary behavior exceeded 18 hours per day.
- Excluded records where sedentary behavior plus activity exceeded 18 hours.
- Excluded records where sedentary behavior plus activity plus cycling exceeded 18 hours.
These procedures, which align with established IPAQ data cleaning protocols, helped to minimize the influence of implausible values on our results. We acknowledge that some variability remains, which may reflect genuine differences in physical activity levels among participants. However, we believe that the steps we took to address outliers provide reasonable assurance that our findings are not unduly influenced by extreme values. The corrected minimum MET-min/week value is 148,5, which is also updated in Table 2.
We acknowledge that we could not definitively determine whether increases in walking were due to the game or pre-existing habits, nor estimate how much was related explicitly to Pokémon GO. We have specified whether psychosocial outcomes were measured with standardized instruments or single-item questions. The additions can be found on page 4, lines 156-161, page 7, lines 209-226, page 8, lines 251-253.
Comment 6: The Discussion provides a broad and well-referenced interpretation of the findings, emphasising the positive impact of Pokémon GO on physical activity and social interaction. However, it often overstates the health implications without sufficiently addressing the study's limitations. For instance, while the increase in ambulation is well described, the authors attribute significant health benefits to this change. Yet, the cross-sectional design and lack of causal analysis do not suport such claims. Did you consider that participants who are already active may have been more inclined to play the game? Moreover, many cited effects (e.g., improved mood, weight loss, social integration) are speculative or based on external literature—were these outcomes measured in your cohort, and if so, how? The concept of “stealth health” is interesting but not operationalised—can you explain how you assessed long-term behaviour change or sustainability of activity beyond short-term enthusiasm? Although the risks (e.g., addiction, inattention, screen time) are acknowledged, they are not analysed in depth relative to the benefits—how did you balance these opposing effects in your interpretation? Finally, the absence of a control group is not mentioned. Yet, it limits the ability to compare behaviour trends or determine whether Pokémon GO use improves health relative to similar users who do not play. A more cautious and methodologically grounded interpretation would enhance the credibility of this section.
Response 6: We thank the reviewer for their critical feedback on the Discussion section. We have toned down the language about health implications and largely emphasized the study's limitations. We have acknowledged that already active participants may have been more inclined to play the game. We have also distinguished between effects that were measured in our cohort and those that are based on external literature. The additions can be found on page 7, lines 209-213, 224-226, page 8, lines 245-247, 251-253, 268-270, 294-298 (ending on page 9), page 9, lines 304-307, 314-321, 330-334.
Comment 7: The Conclusion appropriately summarizes the potential of Pokémon GO to modestly increase physical activity and enhance social well-being. However, it overstates the health benefits without sufficient support from the cross-sectional design. The mention of risks (addiction, injuries) is valid but treated superficially. Did you consider suggesting guidelines for safe and balanced use? A more cautious, evidence-based tone would strengthen the final message.
Response 7: We thank the reviewer for their feedback on the Conclusion. We have used even more cautious language about health benefits and included a brief suggestion of guidelines for safe and balanced use. We have added the sentence: 'Future research should focus on developing guidelines for safe and balanced use of AR games to maximize their potential benefits and minimize their risks.' The additions can be found on page 8, lines 278-283, 294-298 (ending on page 9), page 9, lines 304-307, 314-321, 330-334, page 10, lines 364-366.
Reviewer 3 Report
Comments and Suggestions for Authors
-
Strength of the Introduction and Background
The introduction is comprehensive and well-structured, offering a clear overview of the rise of mobile devices, augmented reality, and their integration into health-promoting activities. The use of Pokémon GO as a case study is well justified, and references [1–14] provide a robust framework. The authors may consider briefly comparing Pokémon GO with other AR-based health tools to enhance the context. -
Clarity and Consistency in Methodology
The methodological section was articulated and appropriately detailed. The use of the IPAQ-long form and a customized questionnaire is methodologically sound. However, the demographic categorization (e.g., "city, >200,000 inhabitants") can be better visualized in the text or as a bar chart to aid readability. Additionally, while the questionnaire design is mentioned, an appendix with actual survey questions would strengthen the transparency and reproducibility. -
Presentation of Results
The tables (particularly Tables 1–3) are comprehensive and statistically informative. For instance, Table 2 detailing MET-min/week by gender and activity type is particularly strong. However, consider including graphical visualizations (e.g., histograms or boxplots) for better engagement and faster comprehension. The distinction between perceived and actual benefits (e.g., self-reported walking vs. MET values) is handled well. -
Discussion and Interpretation
The discussion effectively contextualizes the findings within the broader literature. The incorporation of references [16–55] strengthens credibility. One notable strength is the attention paid to both positive outcomes (increased physical activity, social engagement) and negative effects (screen addiction, sleep disturbances). This might enhance the discussion to include a short segment on how game design could be improved to mitigate risks (e.g., geo-fencing near roads). -
Critical Assessment and Limitations
The limitations of this study are appropriately acknowledged. The lack of a control group and the cross-sectional design are mentioned, but the authors could go further by suggesting how future research might implement randomized controlled trials or longitudinal follow-ups. Potential bias from self-reported data (IPAQ) should also be discussed more explicitly. -
Language and Terminology
The English language is clear, formal, and consistent. There were very few grammatical errors and the terminology used was suitable for a scholarly audience. One suggestion is to maintain consistent terminology for “ambulation” and “walking” unless specifically defined differently. -
Relevance and Appropriateness of References
The citations are extensive and well integrated. This manuscript cites seminal works and recent studies, offering a comprehensive view of the topic. No major missing references were identified. The citations range from foundational public health documents (e.g., WHO guidelines) to domain-specific articles on AR and mobile health interventions.
Author Response
Thank you very much for taking the time to review this manuscript. Please find the detailed responses below and the corresponding revisions and corrections marked in red in the updated version of the manuscript.
Comment 1: Strength of the Introduction and Background
The introduction is comprehensive and well-structured, offering a clear overview of the rise of mobile devices, augmented reality, and their integration into health-promoting activities. The use of Pokémon GO as a case study is well justified, and references [1–14] provide a robust framework. The authors may consider briefly comparing Pokémon GO with other AR-based health tools to enhance the context.
Response 1: We thank the Reviewer for suggesting that we compare Pokémon GO with other AR-based health tools. We have added a sentence to the Introduction mentioning Zombies, Run! and Ingress and highlighting Pokémon GO's unique appeal based on the Pokémon franchise and its focus on collection and social interaction. The addition can be found on page 2, lines 75-82.
Comment 2: Clarity and Consistency in Methodology
The methodological section was articulated and appropriately detailed. The use of the IPAQ-long form and a customized questionnaire is methodologically sound. However, the demographic categorization (e.g., "city, >200,000 inhabitants") can be better visualized in the text or as a bar chart to aid readability. Additionally, while the questionnaire design is mentioned, an appendix with actual survey questions would strengthen the transparency and reproducibility.
Response 2: We thank the Reviewer for this valuable suggestion. We agree that graphical visualisation of demographic categories could improve readability; however, after simulating different layouts, we found that incorporating such figures within the manuscript, while adhering to the journal template, would substantially increase volume and reduce clarity. For this reason, we have retained the tabular presentation as the most concise option.
Regarding custom-survey, we have added an appendix (Appendix A) containing the full list of questions to enhance transparency and reproducibility.
Comment 3: Presentation of Results
The tables (particularly Tables 1–3) are comprehensive and statistically informative. For instance, Table 2 detailing MET-min/week by gender and activity type is particularly strong. However, consider including graphical visualizations (e.g., histograms or boxplots) for better engagement and faster comprehension. The distinction between perceived and actual benefits (e.g., self-reported walking vs. MET values) is handled well.
Response 3: We appreciate the Reviewer's suggestion to include graphical visualizations. We have added a boxplots (Figures 1) showing the distribution of MET values, complementing Table 2 for greater clarity.
Comment 4: Discussion and Interpretation
The discussion effectively contextualizes the findings within the broader literature. The incorporation of references [16–55] strengthens credibility. One notable strength is the attention paid to both positive outcomes (increased physical activity, social engagement) and negative effects (screen addiction, sleep disturbances). This might enhance the discussion to include a short segment on how game design could be improved to mitigate risks (e.g., geo-fencing near roads).
Response 4: We thank the Reviewer for suggesting ways to improve game design to mitigate risks. We have added a paragraph to the Discussion section suggesting specific game design changes, such as geo-fencing and other features (there are singular warnings during game loading or when exceeding speeds of approximately 30km/h). The additions can be found on page 8, lines 278-283, page 9, lines 314-321.
Comment 5: Critical Assessment and Limitations
The limitations of this study are appropriately acknowledged. The lack of a control group and the cross-sectional design are mentioned, but the authors could go further by suggesting how future research might implement randomized controlled trials or longitudinal follow-ups. Potential bias from self-reported data (IPAQ) should also be discussed more explicitly.
Response 5: We appreciate the Reviewer's suggestions for strengthening the discussion and limitations. We have expanded these sections to include suggestions for future research designs (randomized controlled trials, longitudinal follow-ups) and a more explicit discussion of the potential for bias from self-reported data. The additions can be found on page 8, lines 245-247, 263-265 and 268-270 page 9, lines 344-350 (ending on page 10).
Comment 6: Language and Terminology
The English language is clear, formal, and consistent. There were very few grammatical errors and the terminology used was suitable for a scholarly audience. One suggestion is to maintain consistent terminology for “ambulation” and “walking” unless specifically defined differently.
Response 6: We thank the Reviewer for pointing out the need for consistent terminology. We have reviewed the manuscript and ensured consistent use of the term 'walking' throughout, implementing the necessary revisions accordingly.
Comment 7: Relevance and Appropriateness of References
The citations are extensive and well integrated. This manuscript cites seminal works and recent studies, offering a comprehensive view of the topic. No major missing references were identified. The citations range from foundational public health documents (e.g., WHO guidelines) to domain-specific articles on AR and mobile health interventions.
We thank the Reviewer for supportive comment.
Reviewer 4 Report
Comments and Suggestions for Authors
Hello and thank you for the opportunity to reviwer.
In the abstract of the anthropometric characteristics, the measurement methods should be mentioned.
The importance and necessity are not fully stated.
The sensitivity and accuracy of the measurement tools in the questionnaire should be mentioned.
What was the sampling method?
How was the gender difference explained?
In the further discussion, the reasons for the differences and similarities should be discussed.
Resources should be updated.
Author Response
Thank you very much for taking the time to review this manuscript. Please find the detailed responses below and the corresponding revisions and corrections marked in red in the updated version of the manuscript.
Comment 1: In the abstract of the anthropometric characteristics, the measurement methods should be mentioned.
Response 1: We sincerely thank the Reviewer for this valuable observation. We would like to clarify a potential source of misunderstanding. Our study did not involve the collection of anthropometric characteristics (such as height, weight, or body mass index). Instead, we focused on demographic variables (age, place of residence, education, occupation), as presented in Table 1. Consequently, no measurement procedures were required for these variables. If the reviewer believes this is an error and that anthropometric characteristics are in fact present, please let us know the exact location so we can address it. Nonetheless, we added a bracket informing about demographic characteristics in lines 19-20.
Comment 2: The importance and necessity are not fully stated.
Response 2: We thank the Reviewer for pointing out the need to better emphasize the importance and necessity of our study. We have revised the Introduction to more clearly articulate the public health significance of physical activity, the potential of AR games to promote it, and the gap in the literature that our study addresses. The addition can be found on page 2, lines 83-91.
Comment 3: The sensitivity and accuracy of the measurement tools in the questionnaire should be mentioned.
Response 3: We sincerely thank the Reviewer for raising questions about the sensitivity and accuracy of our measurement tools. We have added information about the validity and reliability of the IPAQ and expanded the description of our custom questionnaire to include details about its development and piloting. We acknowledge that the lack of more rigorous psychometric testing of the questionnaire is a limitation. The additions can be found on page 3, lines 109-117 and 125-129.
Comment 4: What was the sampling method?
Response 4: We thank the Reviewer for asking about the sampling method. We have added a sentence to the Methods section clarifying that participants were recruited through convenience sampling via online platforms. The addition can be found on page 3, lines 130-131.
Comment 5: How was the gender difference explained?
Response 5: We thank the Reviewer for this important comment. Gender-related differences have been clarified in the Results and elaborated in the Discussion. We will refine our tools in future studies to better emphasise such differences. The addition can be found on page 4, lines 156-162 and page 7, lines 214-226.
Comment 6: In the further discussion, the reasons for the differences and similarities should be discussed.
Response 6: We appreciate the Reviewer’s suggestion. The Discussion has been expanded to explain both gender differences and similarities: men showed higher work-related activity and greater risk-taking, while women contributed more household activity and individual sports. Despite these distinctions, both genders reported comparable improvements in mood and physical activity, consistent with earlier findings. Future studies will further address mechanisms underlying these gender-related patterns. The addition can be found on page 7, lines 214-226.
Comment 7: Resources should be updated.
Response 7: We thank the Reviewer for suggesting that we update our resources. We have reviewed our reference list and updated it to include the most recent relevant research on this topic.
Round 2
Reviewer 2 Report
Comments and Suggestions for Authors
Dear Authors,
Thank you for your effort in improving the manuscript. Now, the manuscript is engaging and well-written, but I believe that it still needs improvement before being suitable for publication. In particular, these significant aspects should be addressed:
Methodological rigour: The study is based exclusively on self-reported questionnaires, which limits the accuracy of the results. This limitation should be explained more explicitly in the Limitations section. In addition, please consider that your statistical analysis is mainly descriptive or focused on bivariate comparison. It is ok, but it should be underlined and explained why you chose to perform only descriptive statistics. Otherwise, you can improve the statistical analysis by considering multivariate models (e.g., logistic regression or ANOVA with covariates) to control for confounders such as age, gender, and education. In this case, you have to include these new results and discuss them.
Interpretation and caution: The conclusions currently suggest that Pokémon GO broadly promotes a healthy lifestyle. These statements should be rewritten more cautiously, clearly stating that the results are only associative and that causality cannot be inferred due to the cross-sectional design. The conclusions should emphasise that findings are preliminary and highlight the need for longitudinal research with objective measures.
Author Response
We greatly appreciate taking the time to review this manuscript again and provide a second feedback. Please find the detailed responses below and the corresponding revisions and corrections highlighted in yellow in the updated version of the manuscript.
Comment 1: The study is based exclusively on self-reported questionnaires, which limits the accuracy of the results. This limitation should be explained more explicitly in the Limitations section.
Response 1: We thank the Reviewer for this valuable comment. We have revised the Limitations section to explicitly underline that the exclusive reliance on self-reported questionnaires reduces the accuracy of physical activity and psychosocial outcomes. We also noted that future studies should incorporate objective measures such as accelerometers and validated multi-item scales to strengthen validity. The addition can be found on page 9, lines 349-350 and page 10, lines 352-358.
Comment 2: In addition, please consider that your statistical analysis is mainly descriptive or focused on bivariate comparison. It is ok, but it should be underlined and explained why you chose to perform only descriptive statistics.
Otherwise, you can improve the statistical analysis by considering multivariate models (e.g., logistic regression or ANOVA with covariates) to control for confounders such as age, gender, and education. In this case, you have to include these new results and discuss them.
Response 2: We appreciate this constructive suggestion. In the revised Statistical Analysis section, we clarified our rationale for focusing on descriptive and bivariate statistics. Because many variables were nominal/ordinal and several psychosocial outcomes were single-item, multivariable models would have required additional assumptions or transformations not prespecified a priori and could have reduced interpretability. This approach is consistent with common practice in prior Pokémon GO studies cited in our manuscript. We emphasised that our analyses are exploratory and hypothesis-generating, intended to guide future longitudinal research with objective measures. The addition can be found on page 3, lines 140-145.
Comment 3: The conclusions currently suggest that Pokémon GO broadly promotes a healthy lifestyle. These statements should be rewritten more cautiously, clearly stating that the results are only associative and that causality cannot be inferred due to the cross-sectional design.
The conclusions should emphasise that findings are preliminary and highlight the need for longitudinal research with objective measures.
Response 3: We thank the Reviewer for highlighting this important point. The Conclusion section has been revised to adopt a more cautious tone. We now clearly state that the findings are associative, not causal, and emphasise that they are preliminary results from a cross-sectional design. We also underline the need for future longitudinal studies with objective measures to more accurately assess the sustainability and causal impact of augmented reality games on health behaviours. The addition can be found on page 10, lines 363-368 and 371.
Reviewer 4 Report
Comments and Suggestions for Authors
Edited and its can be publish
Author Response
We sincerely thank the Reviewer for their positive assessment and support. We are pleased that the revised version meets the expectations and is now considered suitable for publication.